# Study protocol: Local labour market programs–institutional structures, organizational forms and lived experiences

**Gunilla Olofsdotter** [1†], **Katarina Giritli Nygren** [1]*, **Gunilla Bergström**[2], **Malin Bolin**[1], **Carolina Klockmo** [3], **Sara Nyhlén**[1], **Ida Sjöberg**[1], **Åsa Tjulin**[3]

**1** Department of Humanities and Social Sciences, Mid Sweden University, Sundsvall, Sweden, **2** Research & Development, The Region of Gothenburg, Gothenburg, Sweden, **3** Department of Health Sciences, Mid Sweden University, Sundsvall, Sweden

† Deceased.
* Katarina.giritli-nygren@miun.se

## Abstract

In this programme, we map and examine local labour market programmes (LLMPs) at the municipal level in Sweden. This includes their institutional structure and organisation, as well as the experiences of participants in the programmes, using a longitudinal approach with the aim to improve LLMPs. The long-term goal is to increasing the inclusion of LLMP participants in working life. To answer the programme's questions, data will be collected and analysed within the four work packages. In each work package, a mixed-method approach is applied with a combination of quantitative and qualitative methods. The programme is informed by three overarching general theoretical approached, tying together institutional ethnography, intersectional studies of structural inequalities on different levels, and the role of emotions in everyday work. At the organisational level (WP 1), we will investigate the circumstances under which LLMPs are performed and negotiated by those involved. Here, the internal organisation, activities and methods are the focus. This approach will result in knowledge about the characteristics of these organisations and the factors promoting the inclusion of underrepresented groups in working life. By examining the activities in LLMPs (WP 2), we will be able to determine how their institutional structure differs between regions in Sweden, how the different municipalities work with labour market policy, how they translate national policy into the local context, how they organise their work and which initiatives they choose to adopt. By examining the individual experiences of those who are directly affected by such incentives (WP 3), knowledge and understanding will be obtained of the connections between experiences and labour market policies. This will give important insights into the functioning of local programmes and of the opportunities to create entry into the labour market. Furthermore, in WP4 we will develop and test an effect evaluation of work methods used in LLMPs and their effect on clients' progress over time.

**Data Availability Statement:** Deidentified research data will be made publicly available when the study is completed and published.

**Funding:** Funded studies FORTE, Grant number: STY-2021/0005 forte.se/eng.

**Competing interests:** The study is funded by FORTE, Grant number: STY-2021/0005 forte.se/eng. The funders had no role in study design, data collection and analysis, decision to publish, or preparation of the manuscript."

## Introduction

This research programme aims to map and examine LLMPs, specifically, their institutional structure and organisational forms, as well as experiences of participants in these programmes, using a longitudinal approach. Many people defined as being 'far away' from the labour market are enrolled in local labour market programmes (LLMPs). They constitute an increasing part of the group of long-term unemployed in Sweden and often comprise groups underrepresented in working life.

In our own earlier study [1], LLMP staff report that the target group they meet has changed significantly in recent years. The people the staff meet today are a heterogenous group with very different support needs; there are people from a foreign background whose only difficulty is that they have not mastered the Swedish language, people who suffer from mental illness and depression, people with addictions and/or criminal backgrounds and people with neuropsychiatric disabilities. So far, this group has mainly been understood and regarded as lacking motivation, experience or education and, more recently, the language [2:7, 3,4], and equipping them with the relevant qualifications has been the focus of labour market policies. We argue that a different starting point is required. A recurring theme in the experience of working with these people and the few stories that exist from the people themselves, is that of a complex problem in simple systems that increase the distance from the regular labour market rather than the opposite, and making individuals' progress invisible [1,5]. Recent studies have also illuminated the importance of integrating subjective outcomes in terms of self-assessed health by participants for evaluating the success of such programs [6]. In extreme cases, it may also be the case that effects on objective and subjective outcomes diverge and evaluations based on only on objective measures might only provide a limited depiction of program effects on participants' overall welfare. We believe that promoting processes for people who are far from the labour market requires new perspectives, as well as recognition of the experiences of the people involved and the work that actually takes place within the LLMPs. In addition, it also requires a change in measurement and a follow-up that shows how people approach the labour market instead of just defining them as deviant. Increased knowledge about the participants and the working methods used, as well as new measures that make that work visible, can contribute to real impact evaluations and identification of structural barriers to approaching the labour market. We argue that answers to such questions most likely can be found by exploring the context and content of LLMPs at the municipal level.

### Aim and research questions

The overarching aim of this programme is to contribute to the inclusion of LLMP participants in the labour market. In order to do so, we will (1) map and critically examine the institutional structure, organisational forms and lived experiences of LLMPs using a longitudinal approach and (2) develop and test an effect evaluation of work methods used in LLMPs and their effect on clients' progress over a period of six years. The project is divided into four work packages, through which the following research questions will be answered:

### Local labour market programmes (LLMPs)

Labour market policy is essentially a state responsibility, conducted by the Swedish Public Employment Services (PES). Nevertheless, at the same time, a shift of responsibility in Swedish labour market policy has had major consequences for the municipalities [7,8]. Responsibility for the unemployed is being transferred to the municipalities, and initiatives for the unemployed are being carried out by municipalities. The municipal labour market measures have increased to such an extent that one can now speak of two partially overlapping labour market

policy systems: (1) an economic one that aims to increase the proportion of employed and (2) a strategic one that is about activating, mediating and retraining the unemployed [7,9]. The first system is at the state level and directed at those receiving support from unemployment funds, and the second is at the municipal level and directed at those with income support [10]. This has entailed a financial transfer from state unemployment or health insurance to financial assistance provided by the municipalities' social services [11]. It is in particular individuals 'far away from' the labour market who are enrolled in LLMPs and addressed by the municipally based activation policy. They often do not have alternative means of subsistence and are dependent on the public sector for financial support.

The transference of responsibility is reflected in the increased numbers of municipalities that offer LLMP services. In 2011, around 90% of the country's 290 municipalities had a labour market unit, a number that had increased to around 94% in 2015 [12]. In 2016, the municipalities' labour market activities corresponded to more than a third of the PES's activities [7]. On an annual basis, approximately 110,000 people participate in these programmes [13], most of them directed there by social services, followed by the PES [12]. The broader responsibility for the unemployed placed on municipalities hits the municipal economy hard; municipalities with high unemployment rates spend 2.5 times more money on LLMPs than municipalities with the lowest unemployment rates [12]. Thus, municipalities that already have a strained economic situation due to low tax revenues as a result of high unemployment rates have a much harder time adapting to and coping with the transferred responsibility/their new responsibility.

To understand Swedish unemployment rates–and how the LLMP can be properly assessed and compared–one needs to take regional differences into account. The discrepancy between the number of unemployed and number of vacancies is related to structural changes in terms of regional differences in population, economic growth, industrial structure and community services. These regional differences are also reflected in the number of people enrolled in LLMPs and the number of people with financial support from social services [12].

Despite LLMPs being established in nearly all Swedish municipalities, little is known about how labour market activities are related to the local labour market and how cooperation with local employers is organised, or with what types of industries they cooperate. However, the above-described development indicates the great need for these programmes, even though it is unclear if and how they actually work with regard to people making their own livelihood and becoming established in the labour market [14–16].

## Are LLMPs effective?

There is a lack of knowledge about what LLMPs do and how effective their efforts are, as a consequence LLMPs have been called into question. Although the growing number of LLMPs has entailed a growing focus on the activities that are carried out within them, there is still a lack of research regarding these initiatives [7]. This can be explained, on the one hand, by the fact that it is not a statutory activity for the municipalities, and thus LLMPs do not have an obligation, and on the other hand, by the freedom of municipalities to organise the programmes on the basis of their own local context and needs. As a consequence, comparative statistics are missing.

Nevertheless, since January 2017, the municipalities have been required to provide information on their labour market initiatives on the basis of six common definitions: work preparation efforts, internships, job search activities, courses, study and career guidance, and other municipal labour market initiatives [7]. Using these measures, mappings can give a picture of activities and participants at the municipal level but do not provide information regarding which factors or methods result in jobs or education for the participants.

Differences in activities offered are thus not taken into account, which would have allowed geographical location and/or structural differences in local collaboration and industry to be considered [12]. Moreover, work-oriented rehabilitation and the logics of social policy do not become visible. The discrepancies between LLMPs and the reported information are due to the measures being based on the regular labour market and omission of individual development and progress made towards the labour market [17]. Against that background, we believe that increased knowledge and improved evaluation of the activities in LLMPs are needed.

## Theoretical perspectives

Drawing on our backgrounds in political science, rehabilitation and sociology, the programme is situated within a tradition that considers issues of exclusion and marginalisation in relation to gendered, racialised and classed structures in society. These processes have become evident and central with regard to both institutional structures and everyday work in organisations [18]. Thus, the entire programme will be informed by gender and diversity perspectives. Gender, class and race/ethnicity are neither fixed nor essential categories, but rather social positions that have meaning in and through people's actions in specific contexts and work inequalities [19]. Intersectionality is an important theoretical tool for understanding power relations. It poses questions about how gender, class and race/ethnicity articulate the exercise of power at different levels of society, and how this is linked to social exclusion [20]. We use this as a methodological approach to study how different forms of subordination and privilege are practised. An awareness of how gender is constructed in relation to other categories or power relations will influence both informant selection and our analysis. For example, many women are employed in tasks that are directly connected to people's wellbeing and basic needs in their daily lives and also play a vital role in helping participants in LLMPs to find entry into the labour market. Using the research team's extensive experience researching gender and diversity issues, we take a critical feminist approach to studying policy documents and the gendered, racialised and classed power relations at play [21].

With regard to the theoretical framework, this programme is informed by three overarching general theoretical approaches, tying together institutional structures, organisational forms and individual experiences that will be combined with the specific theories of the different work packages. The three overarching approaches will hold the programme together and will serve to ensure that our research develops methods and theoretical tools to address the challenge of improving the prerequisites and conditions for organising and undertaking work within various forms of employment for people with a weak foothold in the labour market.

First, the programme is placed in the tradition of institutional ethnography (IE), a method of inquiry that starts from the actualities of people's everyday lives and experiences in order to study social relations as they extend beyond the local and the everyday [21]. Following this approach, our analysis will begin with the social relations of everyday activities relating to LLMPs. Through this approach, we will be able to identify 'clues' in the local settings that we will track and map the people in the programmes and track and map the institutions as linked together in chains of activities. In order to understand the ruling relations of LLMPs, institutional ethnography requires us to examine the 'ordinary daily scene', because this 'has an implicit organisation tying each particular local setting to a larger generalised complex of social relations' [22:156]. For instance, recognising how wider political and economic transformations affect modes of work and production in organisations, we argue, can help us understand how these shape identities, bodies and subjectivities.

Second, the programme is placed in the field of intersectional studies of work. Intersectionality is not a method or theory in itself but provides a framework for exploring how structural

inequalities are constructed, how they play out across micro, meso and macro levels of analysis, and what measures can be used to strive for greater equality [23]. The point of using the concept for an explorative research programme like ours is that it helps maintain our focus on the aspects of inequality in all of the different parts of the programme, regardless of whether we are using qualitative or quantitative methods or if we are researching, for example, forms of employment, share of unemployment or experiences of exclusion. Accordingly, the project will be informed by an intersectional perspective that views social categorisations and structures as neither fixed nor essentialist categories, but rather social positions that have meaning in and through people's actions in specific social, spatial and political contexts at different levels of society [20]. Furthermore, the local is understood as not 'only' local, but also as constructed in relation to what are defined as the regional, national and global.

Third, we recognise that the role of emotions in everyday work is in many ways contradictory and problematic. '(E)motions are not "in" either the individual or the social, but produce the very surface and boundaries that allow the individual and the social to be delineated as if they were objects' [24:10]. Following Ahmed's [24] understanding of emotions as performative, we focus not so much on what the emotions at stake 'are' but rather on what they 'do' with regard to the specific working tasks of LLMP workers in relation to the people they work with (e.g. participants, other public servants, potential employers). How are the subjects positioned in relation to each other? How are they influenced by discourses about work and employability, as well as about the people considered to be far from the labour market and the threat they are said to pose to municipal financial stability? Although those working in LLMPs may not be the primary targets of these discourses, they perform their everyday tasks in such a context.

## Study design and methods

The programme has a longitudinal design covering a period of six years. To contribute with knowledge about the role of LLMPs in the inclusion of people 'far from' the labour market, longitudinal analyses are required to identify changes and developments over time. Previous research illuminates, as shown above, that factors at the institutional, organisational and individual levels influence LLMPs' capacity to support the target group. In order to increase our knowledge about these three levels, the programme is organised with a work package that focuses on each level. To answer the programme's questions, data will be collected and analysed within the work packages. In each work package, a mixed-method approach is applied with a combination of quantitative and qualitative methods.

### Sample

The sample design has been chosen in order to create the best possible opportunities for scientific comparisons of LLMPs (conditions and outcomes) while simultaneously obtaining in-depth knowledge of their functioning. Our starting point is an exploration of LLMPs as organisations that are in constant change. Changes are driven both by the organisations themselves, as they conduct non-statutory activities based on municipal self-government, and by the surrounding context. LLMPs are largely governed by the authorities that assign people to them as well as by the ambitions of local politicians, local cooperation between authorities and the willingness of local public and private employers to receive the target group. It is a complex web of relations that may become even more complex due to changes to the PESs in 2022.

To access relevant data, the data collection is designed as a combination of a national population study of LLMP programmes in Swedish municipalities and a comparative case study of institutional structures (WP1), organisational forms (WP2) and individual experiences (WP3)

in two Swedish regions, the Gothenburg region and the region of Västernorrland and a sample of associated municipalities. In addition, a fourth work package (WP4) is to be conducted that focuses on developing an effect evaluation of the impact of working methods on the participants' ability to get closer to the labour market on the basis of the outputs of work packages 1–3.

The two regions, Gothenburg and Västernorrland, are strategically chosen on the basis of 1) structural differences concerning size, regional growth, sociodemographic composition and unemployment and 2) for their close cooperation with regional research and development units. Researchers within the programme have established relations with these municipalities through their work in research and development (R&D) in the chosen regions. They know their history and can use data and results from earlier projects. In both regions, the R&D units are similarly organised and in each regions the R&D unit has organised a regional network consisting of all the municipality's labour market units. As there is a high level of trust between the municipalities and the R&D units, there are good preconditions for cooperation and knowledge transfer between the programme and the R&D units.

One reason behind the choice of the Gothenburg and Västernorrland regions was that the regions would represent regions with different conditions in terms of growth, population development and labour market challenges. The Gothenburg region is one of three metropolitan regions in Sweden. It has long been a growth region, with positive population growth. It consists of 13 municipalities, the smallest having about 13,000 inhabitants and the largest about 583,000. The Västernorrland region, by contrast, is a region of seven municipalities that have been struggling with negative population growth for a long time; several municipalities have experienced closures of companies and community services. The largest municipality has almost 100,000 inhabitants and the smallest has about 9,000. The sample makes it possible to understand how regional conditions affect the work of the labour market units to support individuals' paths to work and education.

The regions also differ in unemployment rates, with average rates of 3.4% for the Gothenburg region and 6.8% for Västernorrland in 2020. The rates have increased during the pandemic, particularly for the Gothenburg region. Despite these differences, the number of long-term unemployed as a proportion of the unemployed is relatively similar for the two regions–50.8% in Gothenburg and 62.4% in Västernorrland–which illustrates that the target group is a problem in both areas.

## Sample of municipalities

The sample of municipalities has been made on the basis of a number of criteria. In accordance with Vikman and Westerberg [11], we note that there are both structural and local factors that contribute to different conditions between LLMPs that need to be considered in a sample. Structural factors refer to differences in municipal residents 'need for interventions and municipalities' ability to manage them financially, e.g. population composition, unemployment, socio-economic sorting, size, demographic dependency ratio and economy (tax power). Local factors refer to factors that the municipalities can influence themselves on the basis of municipal self-government, for example organizational conditions, competence and working methods and level of ambition. Register data from Statistics Sweden and Kolada have been used to identify how municipalities are the same or different based on structural and local factors. These data have been supplemented with informal knowledge about the municipalities based on a long-term practical research through R&D and information obtained via contact persons connected to the LLMP management networks in each region. The selection of municipalities is based on similarity regarding the municipality's relative position within each

region. In the selection process, we have then created a number of municipal pairs with one municipality from each region. It has not been possible to include all municipalities in this process, partly because there are different numbers of municipalities in the two regions, and partly because there are some municipalities that are unique in their kind, for example the city of Gothenburg. The four municipal pairs selected represent different local positions and profiles, which means that we get a breadth in the empirical data. The pairs are equal in respects that are relevant to make visible. By identifying different types of pairs, a selection is given with different extremes and composition of structural and local factors. This type of sample, Most Similar Systems Design, recommends that when you want to study the significance or impact of a phenomenon (e.g. an intervention), you should choose cases that are otherwise as similar to each other in order for the study to be effective [25–27]. A total of eight municipalities are included in the sample, four municipalities in each region.

Type of pairs—based on local and structural factors:

1. The challenged: small municipalities with major socio-economic challenges

2. The active: engaged LLMPs and larger proportion of foreign-born

3. The innovative thinking: holistic thinking in the municipality around the target groups

4. The entrepreneurial: divergent position and culture in the region

The first pair, the challenged ones, are equal in structural factors by having a socio-economic profile that is challenged based on population composition and economy. The second pair—the active and committed ones—are characterized by the fact that they have a relatively high proportion of foreign-born in terms of population. It is also about municipalities with extensive LLMP activities that are at the forefront when it comes to, for example, absorbing new ideas and research. The latter can also be said to be a hallmark of the third pair—the innovators, but this pair stands out in such a way that there is a more pronounced municipality-wide ambition to work for reduced unemployment among vulnerable groups. The fourth pair has structural similarities by having a relatively good labor market. Here one could also talk about an alternative culture or spirit characterized by entrepreneurship. The local work with the target group also differs in that the working methods in the two municipalities are more or less clearly inspired by the so-called Trelleborg model.

## The use of multiple methodologies

The comprehensive methodology in the programme, combining both qualitative and quantitative approaches, allows for a thorough understanding of the LLMPs in Sweden, enabling researchers to assess program effectiveness, participant experiences, and the impact of policies over time. Below we list the methodologies used which are then followed by deeper description in each work package in the next section.

## Quantitative Methods

1. Questionnaire Survey for LLMP Managers:

- Structured questions distributed to LLMP unit managers focusing on organizational structure, staff, turnover, client composition, local agreements, and collaboration with various actors (public and private).

- Complemented by statistics from Kolada regarding budget and unemployment rates.

- Information from municipal websites regarding political governance.

2. Longitudinal Surveys and Questionnaires:

- Conducting surveys and questionnaires with LLMP participants via telephone or video call over multiple years (Years 2–4).

- Focuses on mapping activities, participation levels, changes in programs, and participants' experiences.

3. Cluster Analysis:

- Used for analyzing multifaceted phenomena and identifying groups of observations with similar characteristics. Applied to identify various types of progress among LLMP clients.

4. Regression and Multilevel Analysis:

- Evaluation of identified work methods' effects on client progress using regression and multilevel analysis.

## Qualitative Methods

### 1. Policy Analysis.

- Study using a longer time perspective involving the reconstruction of policy formulation events.

- Collection of empirical material includes policy documents and interviews with political representatives and public administrators.

### 2. Focus Group Interviews.

- Conducting focus group interviews with personnel employed in LLMPs to gain a deeper understanding of interventions, activities, methods used, client interaction, negotiation, implementation, and coordination with various actors for labor market inclusion.

### 3. In-depth Interviews with Participants.

- Initial interviews conducted with LLMP participants from different phases/stages of programs in selected municipalities.

- Follow-up interviews conducted annually until Year 5 of the program.

### 4. Narrative Approach.

- Uses a narrative approach in interviews to understand the personal experience of participating in LLMPs and the impact of policy decisions.

## Integration of Qualitative and Quantitative Methods

**Longitudinal Design Integration.** Integration of data collected from both surveys and in-depth interviews to provide a comprehensive understanding of changes over time, different participant groups, and the impact of the programs.

**Methodological Integration.** Utilizing measurements of work methods and progress gathered from different work packages to conduct effect evaluations and identify patterns over time among different participant groups.

## WP1 institutional structures

The municipalities' increasing responsibility for labour market measures means that the need for municipal initiatives aimed at people who are ascribed the subject position 'far from the labour market' is increasing, but there is a lack of collective knowledge about the content of municipal labour market measures and it is difficult to say how common the initiatives are [15,16,28]. The institutional structure of the LLMPs differs between the different regions in Sweden because the municipalities are fairly free to organise and shape their labour market policy according to what suits them.

The programme starts out with a national population study of all LLMPs in Swedish municipalities to map and to identify whether there are regional types or patterns of LLMP structures (organising, cooperation patterns, governance, political rule, composition of 'participants') in relation to regional labour markets, and if the patterns change over time. As there are no registration data on LLMPs, we conduct this survey to clarify the heterogeneity of LLMPs and how they differ in institutional and organisational factors. We can thereby assess the extent to which our municipalities are representative of the population of LLMPs.

### Research questions

The aim of this work package is to analyse different municipalities' work with labour market policy, how they translate national politics to a local context, how they organise the work and which initiatives they choose to adopt.

**RQ1**: What differences and similarities can be identified in the way LLMPs are structured and organised, how does the pattern change over time and does it have an impact on methods carried out and effects accomplished?

**RQ2**: How are the ruling relations of local labour market policy organised and how are they translated and connected to labour market initiatives at the local level in different municipalities?

### Methods

The mapping will be carried out by a questionnaire distributed to managers of LLMP units. Questions will ask about the units' organisation within the municipality, staff and competence structures, staff turnover, composition of clients, occurrence of local agreements and collaboration with local actors, public and private. The questionnaire is complemented by statistics from Kolada about budget and unemployment rates, and by information from municipal homepages about political government.

For the policy analysis, we will conduct a study that uses a longer time perspective to reconstruct policy formulation events that are the basis for political decisions aimed at organising local labour market units and the labour market initiatives they carry out. This is achieved through the collection of broad empirical material that includes policy documents and interviews with both political representatives and public administrators. We will also include a multilevel perspective and the collaboration between municipalities, county administrative boards and other public authorities.

### Analysis

The national survey will mainly be analysed with descriptive and comparative analyses, but cluster analysis will also be used for exploring varying types of LLMP. Governance relationships can change over time, and this can be studied by exploring the influence of words, metaphors and agency within the discourse of practice–the way people, institutions and practices

exist in relation to one another. In the present study, this is achieved by analysing speech acts as part of more comprehensive governance relationships [21]. We have set out to clarify the connections that frame the work for planning and organising local employment services, and thus we relate the interview material to a larger societal context.

## WP2 Organisational forms

In LLMPs, several concepts exist, such as work ability, employability, skill-enhancing interventions and activation. These concepts all relate to interventions that relate to entering the labour market. However, it is unclear how the staff relate to these concepts and how the content of the interventions relates to the various concepts. Interventions that are visible in national statistics are often activities that aim to increase employability [13,29]. However, the target group taking part in these interventions is described as heterogeneous, and many seem to be suffering from several health and/or social problems. Thus, several interventions might focus on increasing work ability, vocational rehabilitation and contact with various welfare organisations such as health care agencies. The PESs are highly dependent on other welfare organisations to be successful in the individual processes. Sweden has a fragmented and specialised welfare system, which increases the risk that people with complex needs easily fall between the cracks in the system or that the given interventions are not coordinated, sometimes even contradictory. It is necessary for the welfare organisations to cooperate in order for them to be able to carry out their services efficiently and with a high quality [30–32]. There is a need to increase the knowledge about the content of the performed interventions and to examine their contents, the local public and private employers who deliver them and those to whom they are delivered. Here, generating knowledge about these employers' characteristics and making their experiences visible are central to identifying the factors that promote the inclusion of underrepresented groups in working life.

## Research questions

The overall aims of WP2 are to identify how concepts are filled with meaning, understand the circumstances under which LLMPs and interventions are defined, performed and negotiated among personnel in the LLMPs, and understand how municipality units and local public and private employers who have included individuals in their organisations experience the facilitating and hindering factors for labour market inclusion.

**RQ3**: In striving to understand for labour market inclusion within municipalities, broaden the understanding of the content of interventions and activities, and identify if evidence-based methods are used, how are interventions, activities and methods implemented and utilised?

**RQ4**: How are labour market programmes and activities negotiated in cooperation and coordination, both internally within the municipality organisation and externally among key stakeholders (employers, health care, Swedish PES and social insurance agencies), to facilitate labour market inclusion?

**RQ5**: How do employers who have successfully included participants from LLMPs in their organisations understand and experience their role and responsibilities as well as the opportunities for and obstacles to local labour market inclusion?

## Methods

To address RQ3 and RQ4, an explorative cross-case-study design will be undertaken; the cases will be selected on the basis of WP1's mapping of LLMPs. The municipalities will be purposively selected as case studies in approximately 7–10 municipalities. On the basis of the case selection, approximately 1–3 focus groups will be conducted for each case, depending on the size of the municipality and how comprehensive the LLMP unit is. In order to obtain a deeper

understanding of the content in the interventions and activities for labour market inclusion and to identify the methods used, focus group interviews with personnel employed in the labour market programmes will be conducted. The questions in the interviews will focus on the content of the interventions and activities they perform, that is, their methods, how they work with clients, and their negotiation, implementation and organisation of labour market inclusion polices, as well as the cooperation and coordination of key actors identified for labour market inclusion.

## Analysis

Cross-case analysis according to a stepwise-deductive-induction method [33] will be used for the analysis of focus group interviews. The analysis is iterative, through the theoretical lens of work ability and employability [34] and social organisation, which will be theoretically framed by using the concepts of social coordination and bureaucracy [35]. A person's health and well-being are influenced by a spectrum of socioeconomic, cultural, living and working conditions, as well as social and community networks and lifestyle choices. Therefore, to understand the impact of internal and external social relations and their intersectional implications for inclusion in the labour market, we will use the perspective of social determinants for health [36]). In the analysis, we will also explore the kind of support employers need due to the characteristics of the individuals who need labour market inclusion measures.

To address RQ5, an in-depth study design will be undertaken. Data will be based on interviews with internal municipality units of LLMPs and local employers to explore their experiences in relation to successful labour market inclusion. Employers will be recruited internally within the municipality, from different units in the administration, and among local public and private organisations. Approximately 15–20 individual interviews are planned in order to generate sufficient information power [37] but the number can be altered in line with continuous evaluation of data during the research process. To deepen the understanding of the employer perspective on inclusion, the study will undertake a longitudinal approach. That is, data will be collected over time to facilitate an understanding of the nature of social change and the continuity of the process of labour market inclusion. This will give insights for an analysis of successful agency and structural interconnections through micro and macro dimensions and of the support functions needed to facilitate labour market inclusion.

## WP3 individual experiences

In WP3, we are interested in generating a better understanding of the connections between individual experiences and labour market policies, by highlighting how institutional processes shape the everyday worlds in which people live and act. In previous research, the voices of those considered 'far away from the labour market' are underrepresented. Through the collection of narratives of people participating in LLMP activities, valuable insights can be gained, helping us to understand the consequences of policy incentives from the perspective of the lived experiences of those who are directly affected by such incentives [6]. By adopting a similar approach, we want to explore the social organisation of LLMPs from outside institutionalised discourses. The experiences of an individual represent not merely 'a case' but an entry point into the workings of institutions that produce the generalised and abstract characteristics of contemporary societies [22].

## Research questions

The purpose of this work package is to examine the connections between individual experiences and labour market policies, by unfolding and highlighting the lived experiences of enrolment in LLMPs. This will be done by answering the following research questions:

**RQ7**: Who are the LLMP participants? Do the participants differ between municipalities, and if so, how? How does the composition of participants change over time, given, for example, the restructuring of the Swedish PES?

**RQ8**: How do LLMP participants experience and understand their participation in LLMP programmes/activities?

## Methods

Through a combination of qualitative and quantitative methods, this package focuses on who participates in LLMPs and how local labour market inclusion programmes are experienced from the perspective of the participants. Given the programmes' longitudinal design, we will also be able to examine how the clientele changes over time, how the programmes change, how different groups participate in them and how long they remain before entering the regular labour market or being discharged from the programme for other reasons. By combining different methodological approaches–surveys and in-depth interviews–we are able to both map those who participate in different programmes and gather knowledge on how they experience the programmes. Respondents (RQ7) and informants (RQ8) will be recruited through the local LLMP in the sampled municipalities. During the first year of the programme, and when the sampling is done, we will conduct initial interviews with 7–9 LLMP participants from different phases/stages of the programmes, with different needs and from each selected municipality. The data from these interviews will be a first step in familiarising ourselves with the different organisations and their participants. Further interviews will then be conducted on an annual basis until Year 5 of the programme. Use of a narrative approach in these interviews will provide valuable insights into the experience of participating in these programmes and the consequences of political decisions and policy incentives at an individual level. The programmes' longitudinal design will allow us to follow some participants from their enrolment on a programme until their entry into the regular labour market.

In parallel with the interviews, questionnaires will be conducted in Years 2–4 with all participants in the selected municipalities using computer-assisted telephone interviews by phone or video call. The data collected will inform RQ7. By using surveys, we can map activities and participation on a general level and, more specifically, the programmes that different municipalities offer, the activities that participants engage in and whether they experience changes in the programmes during the years of the study.

## Analysis

To answer RQ7, we will mainly use the quantitative data, but the qualitative interviews will also inform the analysis. By conducting a cluster analysis, we will determine the identity of the participants and the groups and issues different municipalities work with, how long different groups of participants are enrolled in the LLMP programme, and what activities they engage in during their time in the programme. By having annual data, we can then compare how the clientele changes over time, how participants move over time and how different policy decisions affect the activities and clienteles.

RQ8 will be answered using the qualitative interviews. The interviews will be analysed using a combination of thematic [38] and narrative analysis [39], adopting an intersectional perspective and focusing on the participants' experiences of the LLMPs. Topics covered will include how they and their needs are met, the obstacles encountered before and during the programme, how the programme assists them in their personal development and how well it prepares them for the labour market.

## WP4 effect evaluation

It has been stated that effect evaluations of behavioural, psychological and social intervention are few in the field of welfare and social services [40]. Effect evaluation is defined as 'research that examines which interventions work best for each client, patient and user and under what circumstances' [40:7]. Like other social science issues, LLMPs as a phenomenon are 'encumbered' by a number of challenges for effect evaluations based on a more scientific logic. One is a lack of access to reliable and comparable data [5]. Testing of causality presupposes that there are independent and dependent variables to analyse [41]. However, there are no validated measures to assess which working methods are used or how participants get closer to the labour market. The data reported to Kolada and the National Board of Health and Welfare as results comprise varying details of education and employment.

Another difficulty is isolating the independent factor, in this case the methods used, from other factors. It is not possible to carry out purely experimental studies of social phenomena with control over other variables that may affect the outcome, with at least one control group and random selection for experiments and control groups to enable separation of effects of the intervention itself from other changes for other reasons [42]. We advocate that quasi-experimental designs be used in order to exclude or at least make alternative explanations less likely [43], while aiming for the systematic accumulation of knowledge about other factors that may affect the results. Nielsen and Simonsen [44] note that evaluations must take into consideration the fact that intervention is surrounded by, or embedded in, contextual factors. By making comparisons and collecting data about contextual factors, we can more easily distinguish what is general and what is unique, and under what circumstances particular methods work best for a client. In this study, data from work packages 1–3 will provide us with such knowledge, which is helpful when interpreting the results.

## Research question

**RQ9:** What is an adequate effect evaluation for the work methods used in LLMPs and for their effect on clients' progress over time?

## Methods

In this work package, measurements of work methods collected in WP2 and measurements of 'progress' in WP3 are used. Clients in LLMPs are often referred to as having complex problems. The measurements of progress are assumed to identify various aspects of this complexity as well as the various forms of progress, since they can involve different factors; it cannot be assumed that a reduction in the extent of a problem equates to progress. Cluster analysis will be used to identify the various types of progress. This involves an explorative pattern analysis aimed at analysing multifaceted phenomena and identifying groups of observations with similar characteristics [45]. In a second step, the types can be validated according to the MOA model, by exploring if some categories (sex, ethnicity, age and disability) are overrepresented (who), and are more frequent in certain municipalities (where). Exploring and control if some categories (sex, ethnicity, age and class) are overrepresented in certain municipalities. The analysis will be repeated yearly and, by analysing clients at different points of time, it will be possible to discern patterns of progress. In addition, we will develop effect evaluations of identified work methods on progress of clients using regression and multilevel analysis. Measurements of methods from WP2 are used as independent variables and measurement of progress from WP3 are used as outcomes. Both clusters of progress and specific variables will be tested because we are striving to develop an effect evaluation. The longitudinal design of the

programme makes it possible to explore if effects occur at different times after exposure and if it differs for different groups.

## Ethical considerations

Ethical approval has been sought from the ethics authority for WP2 and WP3. As these work packages will address areas of health, it means that the programme will process sensitive personal data, which requires ethical testing under the law on human-related research (SFS 2003: 460). The authority has approved the ethics application in two steps. The authority first approved the part of the study which includes the interviews that will inform the questionnaire (Diary number 2022-02320-01). A supplementary ethical application regarding the design of the questionnaire has also been approved (Diary number 2022-05582-02). All work packages included in the programme will follow ethical principles for social sciences and medical research with regard to information, consent, confidentiality and use. More specifically, all participants in the interviews and questionnaires, (such as staff, employers, and the individuals participating in LLMPs activities) will be informed that it is voluntary to participate and that they can, at any moment, choose to suspend their participation without being questioned for their decision. Before they decide to participate they will receive information on the aim of the study, how their information will be handled and they will also have the opportunity to ask questions. All qualitative and quantitative data material will be used for research purposes only, and all material will be securely stored at Mid Sweden University and retained for 10 years, as required for primary research material). Following the law of good research practice (SFS 2019:504) the research team will continuously discuss ethics on programme meetings.

## Discussion

### Summary

In this programme the aim is to map and examine local labour market programmes (LLMPs) at the municipal level, including their institutional structure and organisation, as well as the experiences of participants in the programmes, using a longitudinal approach with the aim to improve LLMPs. At the organisational level, we will investigate the circumstances under which LLMPs are performed and negotiated by those involved. Here, the internal organisation, activities and methods are the focus. This approach will result in knowledge about the characteristics of these organisations and the factors promoting the inclusion of underrepresented groups in working life. By examining the individual experiences of those who are directly affected by such incentives, knowledge and understanding will be obtained of the connections between experiences and labour market policies. This will give important insights into the functioning of local programmes and of the opportunities to create entry into the labour market. By examining the activities in LLMPs, we will be able to determine how their institutional structure differs between regions in Sweden, how the different municipalities work with labour market policy, how they translate national policy into the local context, how they organise their work and which initiatives they choose to adopt.

### Strength of the planned programme

With regard to the strength of this programme, we want to emphasise the following:

- The longitudinal approach, as it makes it possible to follow the effects on LLMPs of political decisions, changed policy rules and regulations, and structural changes in the labour market over time.

- The possibilities of comparison between activities, measures and results of LLMPs for different labour market regions, rates of unemployment and demographic compositions.

- The study design, in which perspectives from institutional, organisational and individual levels will be analysed using quantitative and qualitative methods.

- The fact that the programme has an interdisciplinary competence in sociology, political science and rehabilitation.

Furthermore, the strength of the programme is also reflected in the theoretical, methodological and empirical design. As far as we know, previous studies of LLMPs have not used an intersectional or gender perspective. The absence of such perspectives is also underlined by Paincan and Ulmestig [5]. The methodological approach extends from institutional ethnography, combining narrative interviews with questionnaires and analysis of databases in a longitudinal design. This design makes it possible to explore and compare institutional structures with organisational forms and lived experiences. The uniqueness of the work derives from the inclusion of the participants' own lived experiences as well as the adoption of a longitudinal approach, which will serve to identify contextual factors affecting the organisational forms and composition of participants, its effects on municipal labour market policies in general, and on LLMPs in particular, together with its impact on different groups within LLMPs and their way (back) to a regular labour market.

The interdisciplinary character of the program is also contributing to its strength. The researchers have their background in including sociology, political science, public health and the health sciences and as a consequence, the analytical framework of the programme will comprise a range of approaches, including governance and policy enactment, normalisation and socio-cultural theory, as well as a variety of methodologies including text analysis, in-depth interviews, surveys and comparative and longitudinal analyses. Though diverse in their methodologies and their objects of study, the researchers all share as their key focus the mapping and critical examination of LLMPs, providing the common ground in the programme's deliberations.

The research team will ensure the methodological and theoretical pluralism of the programme, capitalising on the synergies between the researchers in the various work packages and collaborations. The four WPs will be integrated in a shared workshop, in which the researchers will jointly discuss the challenges of the individual work packages and problematise the work of the programme as a whole. The key advantage of this collaborative work will be the enriching encounters between different scientific cultures offered by the research group and its members' cross-disciplinary work.

## Potential risks and challenges

Since this programme is ambitious and takes a comprehensive look at the LLMP-system in Sweden; ranging from the institutional level to the organisational level to the participatory/individual level, this includes a number of risks and challenges outlined in this risk mitigation. The design of the programme, is based on these levels through WP 1, 2 and 3 that will be brought together and inform the policy brief in WP 4. Despite the breadth being one of the programme's major strengths, it also poses a challenge for a programme of this size (nine research questions over four work packages) to bring together the results into coherent and cohesive set of policy prescriptions. As a way of addressing this challenge, we have sought to ensure the programme being stringent and the synergies between the work packages is cherished, at the same time as we can ensure the methodological and theoretical pluralism of the programme. We are therefore planning for shared workshops, in which we jointly will discuss

the challenges of the individual work packages and problematise the work of the programme as a whole and in relation to its overall aim. Through these workshops, and through regulatory meetings in different constellations (the whole research group, some of the work packages, meetings with the reference group etc.), we will be able to discuss emerging findings in the different work packages, what it means for LLMPs across all three levels and the programme as a whole.

A further challenge of the design on the programme, relating to the complexity of the study and the lack of (available) comparative data, and there not being any validated measures to assess which working methods are used or how participants get closer to the labour market. This is the same for anyone doing research on, or evaluation LLMPs or municipal labour market policies in Sweden, as the lack of data makes scientific and qualitative comparisons of LLMPs hard. Therefore, in the case of this programme, we will have to construct many of the measurements that we are interested in, which–as always with quantitative data–comes with a risk of losing the complexity in the programs, methods used, participant composition, outcomes and so on. Complexity that is currently not taken into account in systems like Kolada. This puts pressure on making informed and well thought-trough decisions regarding methods, methodology and operationalisation. However, we believe that, by constructing quantitative measures drawing on the qualitative data from WP 1, 2 and 3 and by taking an explorative approach using cluster analysis, through which we can analyse multifaceted phenomena and identifying groups of observations with similar characteristics [45], we will be able to map and evaluate the complexity within LLMPs in Sweden.

In order to address the challenges and risks outlined above we have designed overall risk mitigation strategies:

## Mitigation plan

Risk 1: Coherently Bringing Together Results from Multiple Work

## Regular Collaborative Workshops

Hold frequent workshops where researchers from each work package can discuss their findings, challenges, and progress. This facilitates knowledge sharing and aids in aligning methodologies and goals.

## Integrated Approach

Encourage communication and collaboration among teams by establishing a common framework or methodology that all work packages can adapt or contribute to. This ensures a cohesive output.

## Continuous Evaluation

Regularly review the progress of each work package to identify potential gaps or inconsistencies. This allows for timely adjustments and alignment.

Risk 2: Complexity of the Study and Lack of Comparative Data

## Constructive Data Measures

Develop and refine measures based on available qualitative data from WP 1, 2, and 3 to ensure they are robust and comprehensive. Use an explorative approach like cluster analysis to identify patterns and nuances in the absence of comparative data.

### Qualitative-Quantitative Integration

Integrate qualitative insights from different work packages into the development of quantitative measures. This ensures a nuanced understanding of LLMPs' complexities in Sweden.

### Thorough Decision-making Process

Establish a rigorous decision-making process for method selection, methodology, and operationalization. Consider the multidimensional aspects of LLMPs to avoid oversimplification.

### Documentation and Review

Keep detailed records of the constructed measures and decisions taken. Regularly review and refine these measures to adapt to emerging complexities.

Overall Mitigation Strategies

### Communication and Feedback Loops

Maintain consistent communication channels between different groups involved in the project. Encourage feedback and discussions to ensure that all perspectives are considered.

### Adaptability and Flexibility

Remain flexible in methodologies and approaches. Adapt to emerging challenges and discoveries to adjust strategies accordingly.

### External Validation

Seek feedback and validation from external experts or stakeholders to ensure the constructed measures and methodologies are robust and aligned with broader perspectives.

By implementing these mitigation strategies, the program can better navigate the challenges outlined, ensuring a more coherent, comprehensive, and robust outcome despite the complexities and limitations mentioned.

### Implications

Nevertheless–despite the limitations mentioned above–, we believe that this research programme proposes a way to collect validated and comparable data that allow LLMPs to be evaluated quantitatively–without local and/or contextual specifics being reduced. Little is known about the effects of Swedish municipal labour market policies in general, the activities carried out in them, or about the participants in and effects of LLMPs. Consequently, our programme will contribute with knowledge regarding how to improve the inclusion of underrepresented groups in working life, and how LLMPs can be effect evaluated and improved.

We argue that a mapping and examination of LLMPs on the municipal level will give insight on the complexity, multiple strategies and activities used. A longitudinal research approach that takes the institutional structure, the organisation as well as experiences of participants into account makes it possible to not only describe what happens in a certain context in a specific time, but also how changes at the societal level, such as political decisions and changed regulations, travel and are transformed in a municipal context over time.

Second, regardless of the election result in 2022, great political reforms are to be expected on the welfare and labour market policy area. Thereby, this programme will be able to follow and evaluate how these reforms effect policy enactment on an institutional, organisational and individual level, informing the policy brief.

Third, by having intersectional and critical theory as a departure point, the programme will fill the knowledge gap left by a lack of intersectional and/or gender perspectives in previous research. By adding such critical perspectives, the intertwining of gender, class, ethnicity, functional variations, etc. will be unfolded, problematised and examined from different levels of analysis. Knowledge about the changing composition of the participants in LLMPs as well as the intersectional effects on process of inclusion will contribute to the development of LLMPs accordingly.

Finally, given the overarching aim of the programme being to contribute to increase the inclusion of underrepresented groups in working life, the programme has societal gains. Therefore, the programme strives to develop knowledge about sustainable measures for enhancing the possibilities for groups such as young people, immigrants and people with functional variations for self-sufficiency, approaching the labour market or studying. By the programme including actors at the institutional and organisational level (e.g. Swedish PES, politicians in municipalities and employees at LLMPs working directly with the participants, as well as local entrepreneurs)throughout the whole research programme and maintenance of continuous contact after the end of the programme, will provide a good basis for continuing work on and implementation of results that are relevant to the development of the services in the local labour market area. Something, that in the long run will increase inclusion on the labour market and in working life.

## Author Contributions

**Writing – original draft:** Gunilla Olofsdotter, Katarina Giritli Nygren, Gunilla Bergström, Malin Bolin, Carolina Klockmo, Sara Nyhlén, Ida Sjöberg, Åsa Tjulin.

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
