## [Decision Letter · Decision Letter 0]

28 Feb 2023

PONE-D-22-31069R1 

It was a mistake, I did not understand the command

PONE-D-22-32841

Study protocol: Local labour market programs - Challenges of and for an inclusive labour market

PLOS ONE

Dear Dr. Olofsdotter,

Thank you for submitting your manuscript to PLOS ONE. After careful consideration, we feel that it has merit but does not fully meet PLOS ONE’s publication criteria as it currently stands. Therefore, we invite you to submit a revised version of the manuscript that addresses the points raised during the review process.

We look forward to receiving your revised manuscript.

Kind regards,

Luu Duc Toan Huynh

Academic Editor

PLOS ONE

Journal Requirements:

"Funded studies FORTE, Grant number: STY-2021/0005 forte.se/eng"     

Additional Editor Comments:

This is a fine work and it could be publishable. However, the literature is not conving without having the comprehensive synthesis of labour market and market evaluation. Please refer to these two studies (encourage) to enhance your arguments:

[a] Litwok, D. (2022). Estimating the Impact of Emergency Assistance on Educational Progress for Low-Income Adults: Experimental and Nonexperimental Evidence. Evaluation Review, 0(0). https://doi.org/10.1177/0193841X221118454

[b] Caliendo, M., & Tübbicke, S. (2022). Do Start-Up Subsidies for the Unemployed Affect Participants’ Well-Being? A Rigorous Look at (Un-)Intended Consequences of Labor Market Policies. Evaluation Review, 46(5), 517–554. https://doi.org/10.1177/0193841X20927237

Reviewers' comments:

Reviewer's Responses to Questions

**Comments to the Author**

1. Does the manuscript provide a valid rationale for the proposed study, with clearly identified and justified research questions?

Reviewer #1: Partly

2. Is the protocol technically sound and planned in a manner that will lead to a meaningful outcome and allow testing the stated hypotheses?

Reviewer #1: Partly

3. Is the methodology feasible and described in sufficient detail to allow the work to be replicable?

Reviewer #1: Yes

4. Have the authors described where all data underlying the findings will be made available when the study is complete?

Reviewer #1: Yes

5. Is the manuscript presented in an intelligible fashion and written in standard English?

Reviewer #1: Yes

6. Review Comments to the Author

You may also provide optional suggestions and comments to authors that they might find helpful in planning their study.

Reviewer #1: This is a study protocol proposing an examination of local labour market programmes (LLMPs) in Sweden. The main goal is to evaluate the performance of LLMPs in promoting inclusion. This would be a significant study to understand Swedish local labour markets, if the following concerns could be addressed properly.

1. The biggest concern I have about this study is that the authors stressed inclusion in their title and introduction. However, this is not evidenced in the research questions. A large part of the study seems to only focus on evaluate the effectiveness of LLMPs, which are relatively irrelevant to "inclusion" metrics. The authors may either stress less of inclusion in their introduction, or make inclusion their key evaluation metric in the four work packages. It is similar for the COVID exogenous shock, which has appeared in many parts of the protocol, but with limited focus in the research questions.

2. While the significance of this study has been justified, as LLMPs broadly serve as a complement of Swedish state function of implementing labour market policies, the significance is argued at national level. The main comparative study covers two regions of Sweden, but I cannot see the selection of these two cases well justified. Have the authors considered all regions in Sweden to make this choice or is this based on feasibility? I would like to see more evidence, and a similar concern is raised to the choice of municipality sample.

3. The research questions are sometimes too broadly defined. Asking "what" questions is inadequate to construct testable hypotheses. I would like to see more detailed plan of variables to be collected and examined, if possible. For example, I would consider Research Question 7 in Work Package 3 is a more testable set of questions. Some of the rest are relatively a bit ambiguous.

4. The authors state this is a longitudinal study. It would be better if the time coverage is also spefified. How will the surveys collect information in the past, and for how long will the study cover?

5. On page 26, there is a typo at the beginning of the risks and challenges section. "ranging from the intitutional lever" which should have been "level".

7. PLOS authors have the option to publish the peer review history of their article (what does this mean?). If published, this will include your full peer review and any attached files.

Reviewer #1: No

---

## [Author Response · Author response to Decision Letter 0]

15 May 2023

Responses to reviews

Journal Requirements:

"Funded studies FORTE, Grant number: STY-2021/0005 forte.se/eng"

Answer to Editor:

The role of Funder is now included in the cover letter: "The funders had no role in study design, data collection and analysis, decision to publish, or preparation of the manuscript."

3. In your Data Availability statement, you have not specified where the minimal data set underlying the results described in your manuscript can be found.

See answer below (4)

4. If you wish to make changes to your Data Availability statement, please describe these changes in your cover letter and we will update your Data Availability statement to reflect the information you provide.

Answer to Editor: 

The paragraph below is now also included in the cover letter

Since this manuscript is a protocol article we do not yet have any data from either interviews or surveys. The data collection will start later this year.

We have established a plan for data handling which means that all data collected will be stored in the university's secure system until any request to access the data.

We follow the ethical guidelines from the ethical review board, which also govern which data can be made available. We strive to make all data available upon request.

Answer to Editor: 

See answer to reviewer below (1). 

Additional Editor Comments:

This is a fine work and it could be publishable. However, the literature is not conving without having the comprehensive synthesis of labour market and market evaluation. Please refer to these two studies (encourage) to enhance your arguments:

[a] Litwok, D. (2022). Estimating the Impact of Emergency Assistance on Educational Progress for Low-Income Adults: Experimental and Nonexperimental Evidence. Evaluation Review, 0(0). https://doi.org/10.1177/0193841X221118454

[b] Caliendo, M., & Tübbicke, S. (2022). Do Start-Up Subsidies for the Unemployed Affect Participants’ Well-Being? A Rigorous Look at (Un-)Intended Consequences of Labor Market Policies. Evaluation Review, 46(5), 517–554. https://doi.org/10.1177/0193841X20927237

Answer to Editor:

Thank you for suggesting these references. We have now added reference to Litewok (2022 on page 16, and Caliendo & Tübbicke (2022) on page 3 and 14. 

Reviewers' comments:

Reviewer's Responses to Questions

Comments to the Author

1. The biggest concern I have about this study is that the authors stressed inclusion in their title and introduction. However, this is not evidenced in the research questions. A large part of the study seems to only focus on evaluate the effectiveness of LLMPs, which are relatively irrelevant to "inclusion" metrics. The authors may either stress less of inclusion in their introduction, or make inclusion their key evaluation metric in the four work packages. It is similar for the COVID exogenous shock, which has appeared in many parts of the protocol, but with limited focus in the research questions.

Answer to reviewer #1:

We understand that this might be confusing since we are not studying inclusion per se, but the long-term goal of our research project is for participants in LLMPs to be included in the labor market. We have tried to make explicit that we focus on the inclusion of LLMP participants perspectives in the measurement of LLMPs effectiveness and that they should have the opportunity to receive the support they need in order for them to be included in the labor market.

We have also removed references to the Covid pandemic because it is not the focus of the study. This also means that the reference list has changed in which the following references has been removed: 

Juranek, S., et al., Labor market effects of COVID-19 in Sweden and its neighbors: Evidence from novel administrative data. NHH Dept. of Business and Management Science Discussion Paper, 2020(2020/8).

Arbetsförmedlingen. Coronakrisen slår hårt mot äldre arbetstagare. AV nyheter 2021. 

SCB. Kraftig försämring på arbetsmarknaden för unga och utrikesfödda. Arbetskraftsundersökningen (AKU) Arbetskraftsundersökningen (AKU) 2020.

2. While the significance of this study has been justified, as LLMPs broadly serve as a complement of Swedish state function of implementing labour market policies, the significance is argued at national level. The main comparative study covers two regions of Sweden, but I cannot see the selection of these two cases well justified. Have the authors considered all regions in Sweden to make this choice or is this based on feasibility? I would like to see more evidence, and a similar concern is raised to the choice of municipality sample.

Answer to reviewer #1:

Thank you for these comments. You address an important issue on the selection of regions and municipalities. One reason behind the choice of regions was that the two regions would represent regions with significant different conditions in, for example, growth, population development and labor market challenges. This means that one of the regions would be one of the Swedish metropolitan regions and the other regions would be a region with more challenging conditions in unemployment rates, population and growth. 

Considering the choices of municipalities in each region we have based the selection on similarities regarding the municipality’s relative position within each region. We have then created four municipal pairs with one municipality from each region. 

3. The research questions are sometimes too broadly defined. Asking "what" questions is inadequate to construct testable hypotheses. I would like to see more detailed plan of variables to be collected and examined, if possible. For example, I would consider Research Question 7 in Work Package 3 is a more testable set of questions. Some of the rest are relatively a bit ambiguous.

Answer to reviewer #1:

Thank you for this comment and we hope that this might be possible for us to consider in following projects. In this program Work Package 3 is more of an explorative study since this is the first study that will investigate the perspective of LLMP participants. The research questions presented for Work Package 3 are the ones developed for the granted project and the questions that have now received ethical permission. This means that we can not change them at this point, and at this point we want to have a more explorative and qualitative approach in Work Package 3.

4. The authors state this is a longitudinal study. It would be better if the time coverage is also specified. How will the surveys collect information in the past, and for how long will the study cover?

Answer to reviewer #1:

We have specified the time coverage of the research program. The program covers six years and information in the past will not be collected although, there are some statistics available at Kolada. Our research will cover the six years period of the project.

5. On page 26, there is a typo at the beginning of the risks and challenges section. "ranging from the intitutional lever" which should have been "level".

Answer to reviewer #1:

Thank you for noticing this typo. This is now changed according to suggestion.

---

## [Decision Letter · Decision Letter 1]

17 Oct 2023

PONE-D-22-32841R1Study protocol: Local labour market programs - Institutional structures, organizational forms and lived experiencesPLOS ONE

Dear Dr. Olofsdotter,

Thank you for submitting your manuscript to PLOS ONE. After careful consideration, we feel that it has merit but does not fully meet PLOS ONE’s publication criteria as it currently stands. Therefore, we invite you to submit a revised version of the manuscript that addresses the points raised during the review process.Please ensure that your decision is justified on PLOS ONE’s publication criteria and not, for example, on novelty or perceived impact.

We look forward to receiving your revised manuscript.

Kind regards,

Eyal Bar-Haim

Academic Editor

PLOS ONE

Journal Requirements:

Additional Editor Comments:

`

The protocol should describe clearly both the quantitative and qualitative methods and also provide some risk mitigation process for the exploratory analysis

Reviewers' comments:

Reviewer's Responses to Questions

**Comments to the Author**

1. Does the manuscript provide a valid rationale for the proposed study, with clearly identified and justified research questions?

Reviewer #1: Yes

2. Is the protocol technically sound and planned in a manner that will lead to a meaningful outcome and allow testing the stated hypotheses?

Reviewer #1: Yes

3. Is the methodology feasible and described in sufficient detail to allow the work to be replicable?

Reviewer #1: Yes

4. Have the authors described where all data underlying the findings will be made available when the study is complete?

Reviewer #1: Yes

5. Is the manuscript presented in an intelligible fashion and written in standard English?

Reviewer #1: Yes

6. Review Comments to the Author

You may also provide optional suggestions and comments to authors that they might find helpful in planning their study.

Reviewer #1: This is a satisfactory revised piece of work that has addressed the concerns from both the editor and the reviewer in the last round. Revisions to the paper are in line with the response letter, and most of the questions have been either reflected in revisions or answered in detail in the letter. Extensive evidence shows that the authors have taken the comments seriously and also re-thought the research question carefully. I think the revised manuscript is of adequate quality to be published. Please refer to the editor's notes if there are additional comments.

7. PLOS authors have the option to publish the peer review history of their article (what does this mean?). If published, this will include your full peer review and any attached files.

Reviewer #1: No

---

## [Author Response · Author response to Decision Letter 1]

5 Dec 2023

Dear Editor:

We are sincerely grateful to the reviewers for their valuable and highly relevant comments. The comments have been really beneficial in assisting us with revisions of the manuscript and we believe they have helped to strengthen the overall quality of the paper. All changes in the manuscript have been highlighted in the manuscript using tracked changes. We hope that the revised manuscript meets your publishing requirements. A more detailed description of the changes is provided below. 

We have acknowledged the responsibility for the integrity of all data collected and analyzed by Gunilla Olofsdotter, who is deceased. We have added the suggested phrasing: 

“Gunilla Olofsdotter passed away before the submission of the final version of this manuscript. Katarina Giritli Nygren accepts responsibility for the integrity and validity of the data collected and analyzed.”

We have added the “†” symbol next to the author’s name in the author list and included a note that this author is deceased.

We confirm that Gunilla Olofsdotters daughter Erica Nordlander, who is a researcher at Gothenburg University (Sweden) can be contacted if the manuscript is accepted for publication: erica.nordlander@socav.gu.se

Response to Additional Editor Comments:

We have reviewed the reference list and ensured that it is complete and correct. 

We have added a section that clarifies the use of both quantitative and qualitative methods, p.10-12. We have also clarified the risk mitigation process and outlined a risk mitigation plan, p. 20-21.

---

## [Editor Report · Decision Letter 2]

10 Dec 2023

Study protocol: Local labour market programs - Institutional structures, organizational forms and lived experiences

PONE-D-22-32841R2

Dear Dr. Giritli Nygren,

We’re pleased to inform you that your manuscript has been judged scientifically suitable for publication and will be formally accepted for publication once it meets all outstanding technical requirements.

Kind regards,

Eyal Bar-Haim

Academic Editor

PLOS ONE
---

## [Editor Report · Acceptance letter]

16 Jan 2024

PONE-D-22-32841R2 

PLOS ONE

Dear Dr. Giritli Nygren, 

I'm pleased to inform you that your manuscript has been deemed suitable for publication in PLOS ONE. Congratulations! Your manuscript is now being handed over to our production team.

Kind regards, 

on behalf of

Dr. Eyal Bar-Haim 

Academic Editor

PLOS ONE